# Analysis of interaction risks of patients with polypharmacy and the pharmacist interventions performed to solve them—A multicenter descriptive study according to medication reviews in Hungarian community pharmacies

**András Szilvay** [ID]*, **Orsolya Somogyi, Annamária Dobszay, Attiláné Meskó, Romána Zelkó** [ID], **Balázs Hankó**

University Pharmacy Department of Pharmacy Administration, Semmelweis University, Budapest, Hungary

* szilvay.andras@pharma.semmelweis-univ.hu

## Abstract

### Objective

The study examined the Drug-Related Problems (DRPs) of patients with polypharmacy in 78 Hungarian community pharmacies, especially the interaction risks in terms of their clinical severity. Also, the objective was to analyze pharmacists' interventions to solve the identified interaction risks.

### Methodology

The research was carried out in the framework of the training of specialist pharmacists at Semmelweis University, with the participation of 78 graduated pharmacists with the collaboration of 98 GPs. A total of 755 patients participated in pharmaceutical counseling which meant a medication review process. DRPs were uniformly categorized and the interventions were recorded by pharmacists, while a detailed analysis of interaction risks was performed by authors.

### Results

A total of 984 DRPs were registered. The most common category of DRPs was the "non-quantitative safety problems" (62.6%). Interaction risk was the most common cause of DRPs (54.0%). The highest proportion of interaction risks were between two prescription drugs (66.7%). In 30.7% of interaction risks' cases, there was not known negative outcome. In contrast, it was recommended to modify the therapy in 14.9% of interaction risks. Acetylsalicylic acid (22.8%), acenocoumarol (17.7%), and diclofenac (13.9%) were the most common active substances which caused serious interaction risks.

**Data Availability Statement:** The data file is available from the Figshare database (DOI: 10.6084/m9.figshare.14687055.v2).

**Funding:** The author(s) received no specific funding for this work.

**Competing interests:** The authors have declared that no competing interests exist.

A total of 599 pharmacist interventions were used to solve the 531 interaction risks. Pharmacists notified the GPs about the problem in 28.4% of cases and they intervened without the GP in 63.1% of cases, most often with patient education (27.4%).

## Conclusion

Medication review by community pharmacists is required for the safe medicine using of patients with polypharmacy, as a significant number of DRPs have been recorded. The incidence of interaction risks stood out. It is essential to develop a pharmaceutical guideline to properly classify the clinical relevance of interaction risks (e.g. according to high-risk active substances) and to increase the collaboration with GPs.

## Introduction

In developed countries, the number of medicines used per person is rising as the population ages and the rate of chronic diseases increases [1]. Between 1995 and 2010, the proportion of people taking more than 5 medicines doubled, while the proportion of people taking more than 10 medicines tripled, affecting 20.8% and 5.8% of the adult population respectively [2]. It can be stated that polypharmacy has become a common phenomenon affecting the elderly [3].

The concept of polypharmacy is defined in many ways in the literature. According to the most widely used definition (as in this manuscript): polypharmacy means the continuous concomitant use of 5 or more drugs [4,5]. A few factors may play a role in the development of polypharmacy, such as poor medical record-keeping, poor transitions of care, use of automated refill systems [6].

Old age and polypharmacy are significant risk factors for the development of drug-related problems (DRPs) [7,8]. According to Pharmaceutical Care Network Europe (PCNE), DRP is defined as "an event or circumstance involving drug therapy that actually or potentially interferes with desired health outcomes" [9]. Drug-drug interaction (DDI) is a common drug-related problem in patients with polypharmacy [6,10–14], due to pharmacodynamic and pharmacokinetic changes caused by aging [15].

DDIs can be grouped into pharmacokinetic and pharmacodynamic interactions. Pharmacokinetic interactions are when one drug affects the concentration of the other medicine through its absorption, distribution, metabolism, or excretion, while pharmacodynamic interaction occurs when the two drugs are taken simultaneously have an additive or opposite effect on the body at the molecular level [16]. In addition to DDIs, there are also drug-disease interactions, as well as drug-dietary (herbal) supplement interactions [17], examples of the latter are interactions caused by St. John's wort with antidepressant effect [18].

Despite the great advances in health sciences and technology interaction risks remain a major problem for health systems and have been increasing over the past decade [19–22]. Their prevalence is between 16% and 91% from research carried out with different methodologies and populations [21,23–25]. DDIs appear at all levels of the healthcare system, especially in hospitalized patients [26]. Previous research has shown that clinically relevant interactions occur between 0.7 and 39 times per 1,000 prescriptions [27,28], so 3.8–9.3% of the population is at risk [27,29].

DDIs carry a serious health risk: by reducing (or enhance by an additive effect) the effectiveness of therapy, they increase morbidity and mortality [30] and increase the risk of hospital admission, which also places a financial burden on the healthcare system [31,32].

Due to all this, it is essential to identify and adequately evaluate potential DDIs [33]. The healthcare professional needs to be able to distinguish clinically relevant interaction risks from insignificant ones, which requires an appropriately critical approach. Pharmacists' medication review and medication reconciliation have a crucial role in this [34,35], especially in community pharmacies, which are the last in the healthcare provider chain.

In Hungary, pharmacists receive a degree after five years at university. In the first two years, basic science knowledge (mathematics, chemistry, biology, botany, etc.) is taught to students, while in the second and third years, they study basic medical knowledge (cell biology, biochemistry, physiology, etc.). This knowledge is the basis of pharmaceutical subjects in the 4th and 5th years, including the study of pharmaceutical care as a separate subject for one semester, during which they get acquainted with the most important therapeutic situations in community pharmacies (antibiotic use, bandages, asthma care, diabetes care, etc.), mainly with theoretical education.

The health care institutions that cover Hungary most evenly are the community pharmacies operating as part of the primary care. The majority of patients visit pharmacies for two reasons: 1) to get a drug prescribed by a general practitioner or a specialist; 2) to seek advice on relieving their mild symptoms. During a consultation, pharmacists dispense the prescribed drug or recommend an over-the-counter (OTC) medication for the patient's symptoms. In Hungary, community pharmacists have a statutory task of detecting clinically significant interaction risks in community pharmacies within the framework of pharmaceutical care [36]. A study looked at the incidence of 39 potentially dangerous interactions: these each occurred in 0–335.89/100,000 prescriptions per year [37]. Previous research showed, that the interaction risk was the most common cause of DRPs detected in Hungarian community pharmacies [38], but at present, little real-life information is known on the clinical relevance of interaction risks, and pharmacist interventions to prevent them.

The study aimed to examine the interaction risks of patients with polypharmacy entering community pharmacies in terms of their incidence (also relative to all DRPs), nature, and clinical severity. Also, the objective was to analyze and enhance the practical effectiveness of pharmacist interventions to counter the identified risks and prepare a procedure for the uniform handling of drug interactions.

## Methods

### Description of the project

**Study design.**  The study was a multicenter descriptive study carried out in Hungarian community pharmacies.

**Study duration.**  The project was carried out between October 2017 and March 2018.

**Study population.**  The research was carried out with the participation of graduated pharmacists (they did not receive monetary compensation). No randomization was used in the selection of participating pharmacists, in Hungarian community pharmacies serving as their workplace, which were accredited pharmacies at Semmelweis University.

The enrollment of the patients was made by pharmacists in community pharmacies using convenience sample technique. Every pharmacist had to invite nearly 10 patients to participate in the project. The survey involved volunteers over the age of 18 with polypharmacy (using 5 or more drugs continuously [4,5]) who got their medications monthly. Patients were invited to participate based on whether a detailed medication review by pharmacists was warranted based on the professional opinion of the patients' general practitioners (GPs).

For the project to be successful, each pharmacist had to try to include at least 1 GP in the research whose patients receive their medicines at that pharmacy.

## Study process

The research was carried out in the framework of the training of specialist pharmacists at Semmelweis University. At the beginning of the project, pharmacists received a one-day course at Semmelweis University, during which participating pharmacists were introduced to the detailed goals, implementation steps, and professional content to be used. To implement the project, a statutory professional guideline [39], as well as the methodological bases of „Metabolic Syndrome Pharmacological Care Program 2.0." (see the classification of drug-related problems) [45], were used by pharmacists, which professional materials were available to all pharmacists before the presented study. The tools to ensure the practical use of these materials (e.g. tables, questionnaires) and the procedure for documentation have been developed by the authors of the manuscript based on the experience of previous pilot projects [38].

The GPs involved received a written summary of the project implementation steps, and the content of the one-day training was available online with the help of the cooperating pharmacists. In addition to assisting pharmacists in involving patients, GPs provided help in resolving DRPs if they were approached by participating pharmacists with the problem.

The patients enrolled had to participate in pharmacist consultations monthly for 3 months, in which they took part in a medication review by the pharmacists participating in the study. During the consultations, pharmacists carried out medication reviews on the patients' entire drug list (Fig 1). If no problems were found, they dispensed the medications. In the event of a DRP detected by the pharmacist, a pharmacist intervention was carried out: the GP of the patient was informed, or the problem was solved independently, on the pharmacist's authority. The DRPs detected were categorized by pharmacists and then recorded in an electronic data table prepared by the authors. At the same time, pharmacist intervention to solve the DRP was also recorded. The system used to classify the detected DRPs [41] divides the problems into six groups (Table 1). In addition to the six groups, the classification system also identifies the possible underlying causes of a particular drug-related problem. The classification system used was chosen and used in the research based on previous successful Hungarian projects [38,40].

The first consultation was followed by at least two further face-to-face meetings monthly during the project (Fig 1).

## Data processing

**Quantitative and qualitative analysis of interaction risks.**   From the data collected by pharmacists, the authors determined the amount of DRPs and interaction risks per capita and the percentage of each DRP category and underlying cause in proportion to the total DRP.

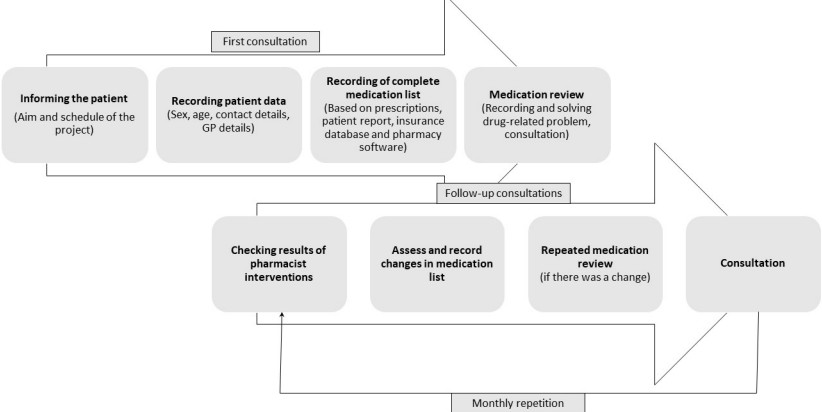

**Fig 1. The procedure of pharmacist consultations during the first and the monthly follow-up sessions.**

**Table 1. Drug-Related Problem (DRP) classification and their underlying cause [41].**

| | | Drug-related problem | Underlying cause |
|---|---|---|---|
| **Necessity** | **DRP1** | Untreated health problem. The patient suffers from a health problem as a consequence of not receiving the medicine that he/she needs. | Medication is necessary (lack of the required medication) |
| | **DRP2** | Effect of unnecessary medicine. The patient suffers from a health problem as a consequence of receiving the medicine that he/she does not need. | Unnecessary taken drug |
| | | | Multiple drug use from the same pharmacological category |
| **Effectiveness** | **DRP3** | Non-quantitative* ineffectiveness. The patient suffers from a health problem associated with the non-quantitative ineffectiveness of the medication. | Improper medication choice |
| | | | Non-adherence |
| | **DRP4** | Quantitative** ineffectiveness. The patient suffers from a health problem associated with the quantitative ineffectiveness of the medication. | Improper dosage |
| **Safety** | **DRP5** | Non-quantitative* safety problem. The patient suffers from a health problem associated with a non-quantitative safety problem of the medication. | Interaction |
| | | | Side effects |
| | **DRP6** | Quantitative** safety problem. The patient suffers from a health problem associated with a quantitative safety problem of the medication | Improper dosage |
| | | Other | |

*Non-quantitative DRP: The DRP is drug-related, it does not depend on the magnitude of an effect.

**Quantitative DRP: The DRP depends on the magnitude of an effect.

During the medication review, in addition to the fact of interaction risks, pharmacists wrote down the active substances involved in the particular interactions and recorded the pharmacist intervention(s). We determined the incidence rate of each active substance. Interaction risks were grouped from two perspectives:

1. Prescribing and dispensing category of the medicine [42], that is, which healthcare professional is competent to provide the necessary intervention and a complete solution (Table 2);

2. The clinical risk of the interaction based on the UpToDate Lexicomp® interaction database [43] (Table 3).

**Comparative statistical analysis of interaction risks.** The frequency of interaction risks considered to be clinically relevant was compared by two demographic aspects (gender, age). We examined the incidence of Grade C, D, or X interaction risks separately in men and women, and in patients under or more than 65 years. In addition, the combined incidence of Grade C, D, and X interaction risks was also examined for both aspects (gender, age).

**Table 2. Grouping of interactions based on the prescribing and dispensing category of the medicine, and on which healthcare professional is competent to provide the necessary intervention and a complete solution.**

| Notation | Explanation | Competent healthcare professional |
|---|---|---|
| **Rx-Rx** | Interaction risk between a prescription drug and another prescription drug detected by a pharmacist. | In most cases, a collaboration between GP and pharmacist is necessary |
| **Rx-OTC** | Interaction risk between a prescription drug and an over-the-counter medicine detected by a pharmacist. | Pharmacist—sometimes in collaboration with a GP |
| **Rx/ OTC-Other** | Interaction risk between prescription or over-the-counter drugs and other products detected by a pharmacist. | Pharmacist—sometimes in collaboration with a GP |
| **OTC-OTC** | Interaction risk between an over-the-counter medicine and another over-the-counter medicine detected by a pharmacist. | Pharmacist |

Rx: Prescription drug; OTC: Over-the-counter drug; Other: Other products (e.g. dietary supplements). Note: The table is based on the classifications in force in Hungary at the time of data processing [42].

**Table 3. Grouping of interactions from a clinical risk perspective based on UptoDate Lexicomp® interaction database [43].**

| | Clinical Risk Grade | Definition |
|---|---|---|
| A | No Known Interaction | "Data have not demonstrated either pharmacodynamic or pharmacokinetic interactions between the specified agents." |
| B | No Action Needed | "Data demonstrate that the specified agents may interact with each other, but there is little to no evidence of clinical concern resulting from their concomitant use." |
| C | Monitor Therapy | "Data demonstrate that the specified agents may interact with each other in a clinically significant manner. The benefits of concomitant use of these two medications usually outweigh the risks. An appropriate monitoring plan should be implemented to identify potential negative effects. Dosage adjustments of one or both agents may be needed in a minority of patients." |
| D | Consider Therapy Modification | "Data demonstrate that the two medications may interact with each other in a clinically significant manner. A patient-specific assessment must be conducted to determine whether the benefits of concomitant therapy outweigh the risks. Specific actions must be taken in order to realize the benefits and/or minimize the toxicity resulting from concomitant use of the agents. These actions may include aggressive monitoring, empiric dose changes, choosing alternative agents." |
| X | Avoid combination | "Data demonstrate that the specified agents may interact with each other in a clinically significant manner. The risks associated with concomitant use of these agents usually outweigh the benefits. These agents are generally considered contraindicated." |

**Analysis of pharmacist interventions to solve interaction risks.** Interventions to address the DRPs revealed during the medication review were recorded by pharmacists on the electronic data collection sheet. The pharmacist interventions described to address the identified interactions were grouped according to Table 4. More than one intervention may have been necessary to solve one DRP.

During the project, pharmacists used a table summarizing the full medication of the patient, which could indicate the current problem (eg interaction and its severity, and related warning), in addition, pharmacists provided patients with written leaflets summarizing the general rules of medication and placed a counseling poster in pharmacies.

**Statistical analysis.** The statistical analysis of the data was carried out using the SPSS 20.0. software (SPSS Inc., Chicago, IL, USA). Based on the descriptive statistics, in the case of Grade C, D, or X interaction risks (separately, and together), the relationship between "one or more interactions" versus "no interactions" by gender and age was examined by the chi-square test. The significance level was 5%.

## Ethics approval

In Hungary according to Regulation No 44/2004 MoHSFA and Act XLVII of 1997, pharmacies did not need to be individually ethically licensed, because the service complies with statutory regulations, and pharmacies are legally entitled to perform such activities [36,39,44,45]. However, the research has been accepted by Semmelweis University Regional and Institutional Committee of Science and Research Ethics (SE RKEB: 110/2021). Verbal informed consent was obtained from all participants in the pharmacies (GDPR decree not yet enacted); no written consent was required according to the Act CLIV of 1997 on Health (noninvasive pharmaceutical service and questionnaire survey) [46]. The investigation was a free service of pharmacies with operating licenses. The patients involved voluntarily participated in the process. Patients participating in the project received verbal information following the national regulations mentioned above. Qualified pharmacists conducted the project. The data were handled by pharmacy and health data management according to Act XLVII of 1997. Data

**Table 4. Definitions of pharmacist interventions to solve drug-related problems.**

| Name of pharmacist intervention | Definition |
|---|---|
| Dosage change | DRP has been resolved by a change in the dosage regimen of a given medicine recommended by the pharmacist. In the case of interaction risks, e.g., pharmacokinetic interactions, this may mean a change in the moment of the day when the drug is administrated. |
| Drug recommendation | DRP has been solved using a new medicine (OTC or other product) recommended by the pharmacist. When there was an interaction risk with a new medicine chosen by the patient in the event of an untreated health problem (DRP1), the pharmacist was able to recommend a safe alternative. |
| Drug replacement | DRP has been resolved by replacing a particular medicine (OTC or other product) with another medicine (OTC or other product) on the pharmacist's recommendation. The interaction risk can be eliminated by a non-prescription replacement of one of the OTC or other products. |
| Education | DRP was solved only with the advice of the patient's pharmacist, without changing the medication regimen. In the case of interaction risk, this means an awareness-raising activity about the fact of the risk, e.g., in the case of Grade C interactions. The patient should be aware of the unintended effect that may occur when taking interacting agents. |
| Helping with device | DRP has been resolved using a device provided by a pharmacist (e.g., a medicine dispenser box, pill cutter). For example, in the case of a pharmacokinetic interaction risk, when time segregation of taking interacting agents is essential according to the times of the day. |
| Not happened | The solution of the DRP would have required intervention by the pharmacist, but this did not happen. In the case of interaction risks, this can be interpreted in the case of Grade C, D, or X risks (Table 3). |
| Not necessary | No intervention by the pharmacist was necessary to resolve the DRP. In the case of interaction risk, although recorded, the pharmacist did not consider the problem to be clinically significant, e.g., interactions classified as Grade A and B. |
| Sending to the doctor | The DRP was solved by sending the patient to the GP, but the pharmacist did not specifically report the problem to the GP. In the case of interaction risk, prescribed medication was involved, but it was not urgent to notify the GP, e.g., for Grade C or D risks. |
| Stop drug | DRP has been solved by stopping taking a particular medicine (OTC or other product) on the pharmacist's recommendation. In interaction, it was possible to stop taking a medication because it was an OTC or other product that the patient did not need. |
| Notification of the GP | The DRP was solved with the patient's GP's help, with the pharmacist notifying the GP (e.g., in person, by phone). In the case of interaction risk, it was important to indicate it to the GP as a matter of urgency, e.g., for a Grade D, but mainly for Grade X risk. |
| No data | No data was available on the intervention to resolve the DRP. |

Interaction Grade A, B, C, D, and X are defined in Table 3.

were transmitted without personal information to process the results. The personal and health data of the patients included in the study were kept anonymous.

The project was implemented with the support and cooperation of the National Health Development Institute's Primary Care Directorate (NHDI-PCD). The professional method provided during the project was a material agreed with the NHDI-PCD.

# Results

## Characteristics of participating pharmacies and patients

The research included 78 pharmacists, 98 GPs, and 755 patients with polypharmacy. The project covered 15 from 20 counties (including Budapest, the capital of Hungary), 35 settlement,

**Table 5. Characteristics of patients surveyed.**

| | | |
|---|---|---|
| **Sex** | Male: | 39.2% |
| | Female: | 60.8% |
| **Age** | < 65 years: | 32.2% |
| | ≥ 65 years: | 67.8% |
| | Min: | 23 years |
| | Max: | 101 years |
| | Average ± S.D: | 69.2 ± 11.2 years |
| **The average number of medicines ± S.D** | Sum: | 9.3 ± 3.3 |
| | Rx: | 7.7 ± 2.8 |
| | OTC: | 1.1 ± 1.2 |
| | Other product: | 0.5 ± 0.9 |

Rx: Prescription drug; OTC: Over-the-counter drug; Other: Other products (e.g. dietary supplements); S.D: Standard deviation; n = 755.

so the survey can be considered to have almost nationwide coverage. The highest proportion of community pharmacies were from Budapest (44%), while 21% from small town (population: 10,000–25,000), 15% from medium town (25,000–100,000 people), 10% from village (<10,000) and 6% from large town (100,000–250,000). Characteristics of patients surveyed in the project are included in Table 5.

## Results of the comprehensive analysis of drug-related problems and underlying causes

A total of 984 DRPs (1.3 DRPs per patient) were registered during the survey. The vast majority of DRPs were non-quantitative safety problems (DRP5; 62.6%). The second most common was non-quantitative ineffectiveness (DRP3; 11.6%). In 8.2% of cases, untreated health problem (DRP1) was detected and the effect of unnecessary medicine (DRP2) was detected with the same frequency. Quantitative ineffectiveness (DRP4; 5.0%) and quantitative safety problems (DRP6; 4.4%) were detected with the lowest frequency. Uncategorized, "Other" problems did not occur. Looking at the underlying causes of DRPs, we found that interaction was by far the most common cause (54.0%), with a total of 531 interaction risks (0.7 per patient) were found by participating pharmacists. The distribution and order of occurrence of the underlying causes are shown in Fig 2.

## Results of detailed analyzes of interaction risks

**Active substances participating in interaction risks.** A total of 135 active substances were identified in 531 interaction risks. The five most common of these were amlodipine (13.7% of interactions), perindopril (13.6%), acetylsalicylic acid (11.7%), metformin (9.8%) and bisoprolol (9.2%).

**Analysis of interaction risks based on the prescription and dispensing category of the drugs.** The distribution of interaction risks is grouped by prescribing and dispensing category of the interacting medicines shown in Fig 3. The highest proportion of interactions were between two prescription drugs (Rx-Rx) that could be solved in most cases in collaboration with GP (66.7%). The incidence of interaction risk types (Rx-OTC, OTC-OTC, Rx/OTC--Other) that can be solved primarily by pharmacists was 31.1%. The majority were interactions between prescription and over-the-counter medications (25.8%).

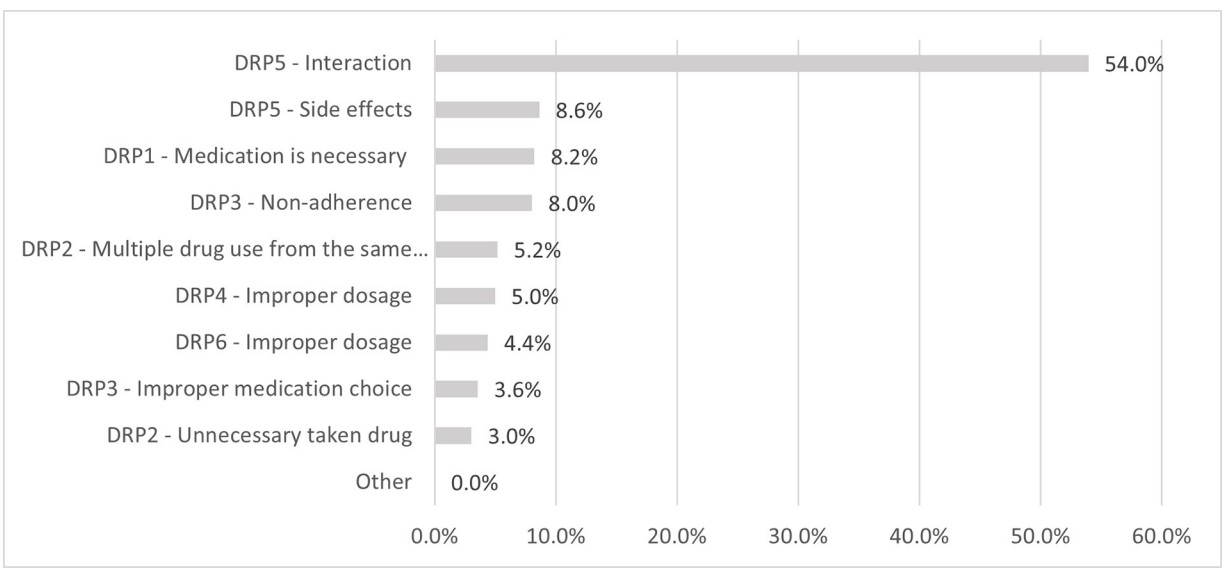

**Fig 2. The distribution and order of occurrence of the underlying causes of drug-related problems.** DRP: Drug-Related Problem; n(DRP) = 984.

**Analysis of interactions and major active substances involved, grouped by clinical risk.** According to the risk classification of the interactions, 42.0% of the cases (would have) made it necessary to monitor the therapy (Grade C). In 30.7% of cases, although the pharmacist suspected a clinically relevant problem between two active substances, according to the UpToDate Lexicomp® database, there was no known negative outcome using the two substances together (Grade A) moreover, in 6.4% of cases, although there was an interaction, no

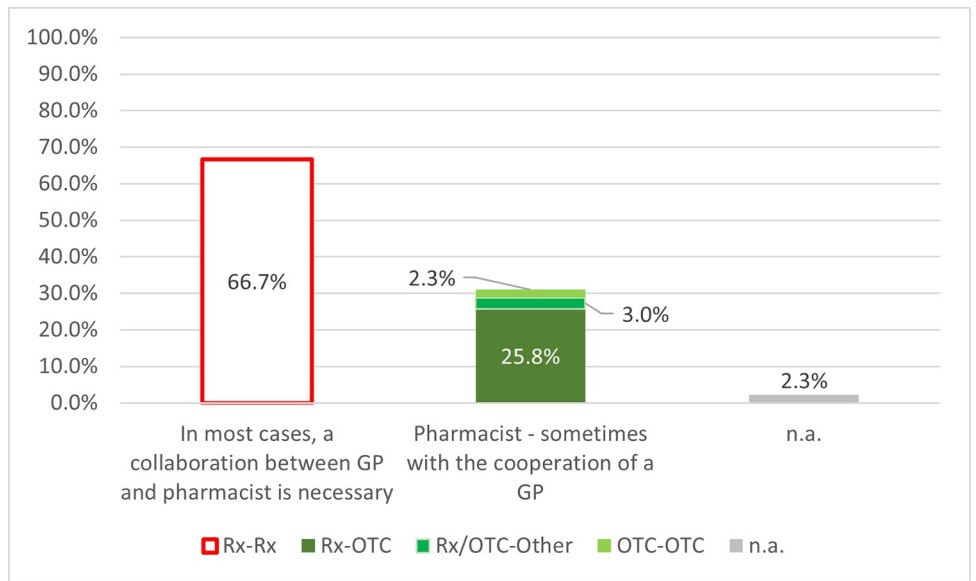

**Fig 3. Frequency of interaction risks grouped by the competent healthcare professional providing the necessary intervention and a complete solution, and by the prescribing and dispensing category of the drugs involved.** Rx: Prescription drug; OTC: Over-the-counter medicine; Other: Other products (e.g. dietary supplements); n.a.: Not available; n = 531.

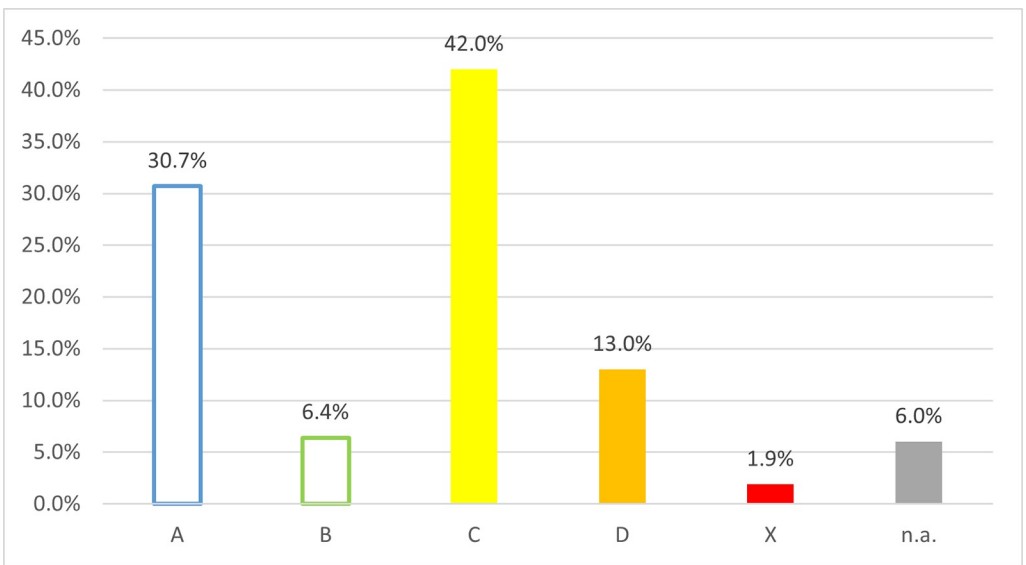

**Fig 4. Distribution of interactions by UpToDate Lexiomp® risk classification grades [43].** A: No known interaction, B: No action needed, C: Monitor therapy, D: Consider therapy modification, X: Avoid combination; n.a.: Not available; n = 531.

further action was required (Grade B). In contrast, in 13.0% of interactions it was (would have been) recommended modifying therapy (Grade D) and in 1.9% the cessation of interaction was (would have been) possible only by the complete elimination of one active substance (Grade X). 6.0% of the interactions could not be categorized because one of the participants was not in the UpToDate Lexicomp® database (Fig 4).

Grade A or Grade B interactions were caused by 77 active substances. The most common active substances (more than 10.0% of the Grade A or Grade B interactions) were amlodipine (32.0%), bisoprolol (17.8%) perindopril (12.7%), and metformin (11.2%). Of these, three agents have highly associated with Grade A or Grade B interaction by participating pharmacists: 71.2% of amlodipine cases; 71.4% of bisoprolol cases; while 42.3% of metformin cases were not clinically relevant. Compared to them, perindopril had a lower rate of Grade A or Grade B interactions (22.2%) (Fig 5A).

By examining the active substances that cause serious (Grade D or Grade X) interactions, we found that acetylsalicylic acid (22.8%), acenocoumarol (17.7%), and diclofenac (13.9%) were the most common (more than 10.0% of the D and X interactions) of the approximately 56 active substances causing such interactions. Of these active substances, a high percentage of acenocoumarol interactions belonged to Grade D or X (60.9%), but the ratio of serious interaction of diclofenac (36.7%) and acetylsalicylic acid (29.0%) is also high (Fig 5B). The Grade D or X interaction pairs of these three agents are shown in Table 6.

**Results of comparative statistical analysis of interaction risks.** In comparative studies by gender, the incidence of Grade C, D, or X interaction risks was examined separately and then aggregated (C + D + X). No significant difference was found between men and women in either case (p>0.05). Examining the age groups, we found that there was a more frequent Grade C interaction risk in the age group 65 years or older, with a significant difference (p = 0.05), while no significant difference was found for Grade D or X interaction risks. Looking at the combined incidence of Grade C, D, and X interaction risks, it can be assumed in professional practice that the older ($\geq$ 65 years) age group is more likely to have clinically relevant interaction risks than the younger age group (p = 0.076, close to the significance limit).

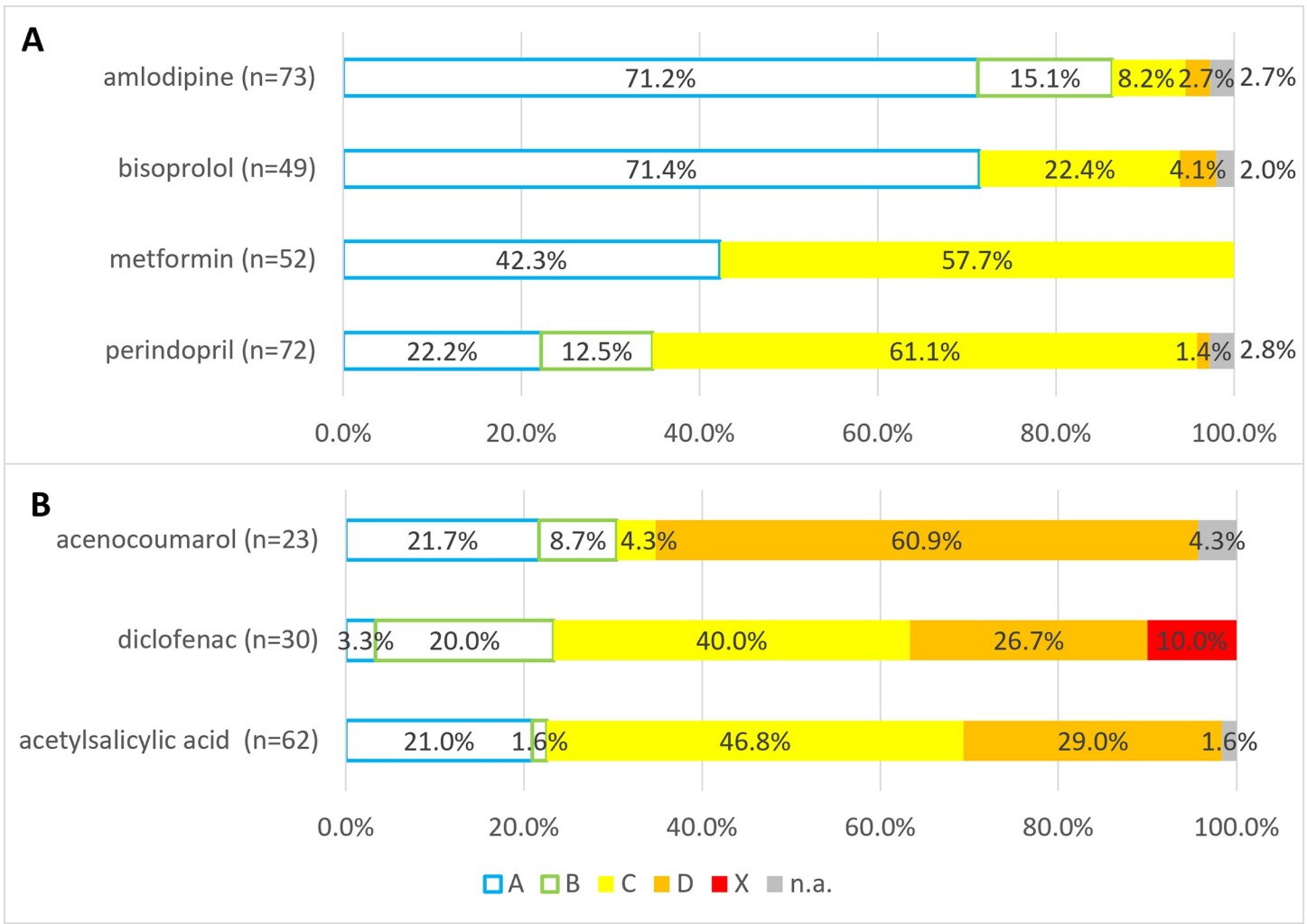

**Fig 5. Distribution of interactions by clinical risk caused by active substances causing the most Grade A/B or Grade D/X interactions.** 5/A: Distribution of interactions by clinical risk caused by active substances causing the most Grade A/B interactions. 5/B: Distribution of interactions by clinical risk caused by active substances causing the most Grade D/X interactions. A: No known interaction; B: No action needed; C: Monitor therapy; D: Consider therapy modification; X: Avoid combination; n.a.: Not available.

**Table 6. Grade D or X interaction pairs of the three most common D or X interacting agents (acenocoumarol, acetylsalicylic acid, and diclofenac).**

| Active substances causing Grade D or X interaction with <u>acenocoumarol</u> | Active substances causing Grade D or X interaction with <u>acetylsalicylic acid</u> | Active substances causing Grade D or X interaction with <u>diclofenac</u> |
|---|---|---|
| 5-aminosalicylic acid | aceclofenac | acetylsalicylic acid |
| acetylsalicylic acid | acemetacin | metamizole |
| allopurinol | acenocoumarol | furosemide |
| garlic | apixaban | aceclofenac |
| Ginkgo biloba | diclofenac | heparin |
| ginseng | enoxaparin | nimesulide |
| ibuprofen | garlic | warfarin |
| metamizole | Ginkgo biloba | |
| piroxicam | metamizole | |
| | nimesulide | |

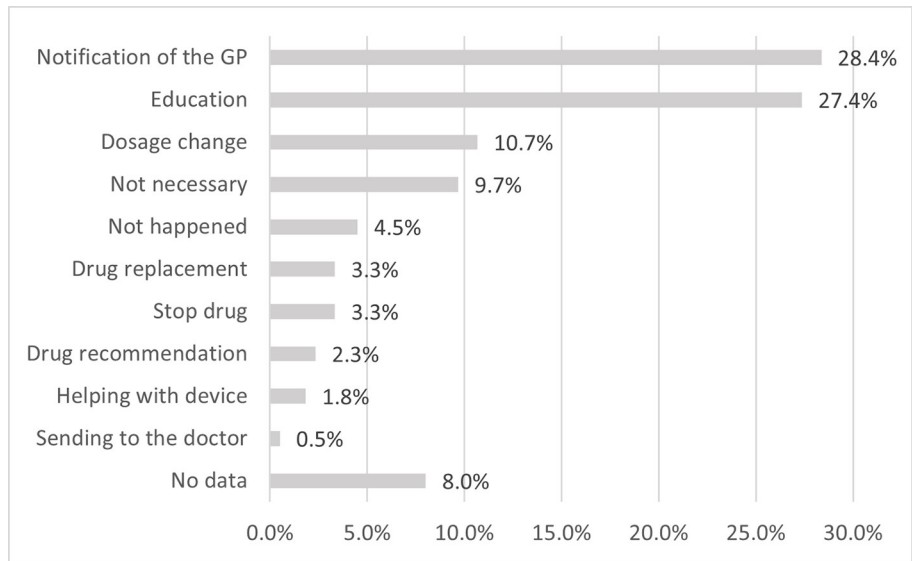

**Fig 6. The incidence of pharmacist interventions to solve interaction risks, in order of frequency.** GP: General practitioner; n = 599.

### Results of analysis of pharmacist interventions to solve interaction risks

A total of 599 pharmacist interventions (Table 4) were used to solve the 531 interaction risks identified by pharmacists. Pharmacists notified the GP about the problem in 28.4% of cases, compared with nearly two-thirds (63.1%) solved the problem without the GP's involvement. Based on pharmacist reports GPs were notified mainly by telephone or in-person, pharmacists did not prefer written contact.

In most of the latter solutions, "education" (27.4%) and "dosage change" were used (10.7%) (Fig 6). In the education of the patients, the information materials developed for the project were used by the pharmacists.

## Discussion

The results of our research confirm the importance of the special attention given to patients with polypharmacy. A significant part of DRPs can be detected with the help of pharmacists working in community pharmacies, helping the workload of GP care in Hungary. Some of the detected DRPs can be solved independently, and others with the help of a GP.

It can be stated, when GPs considered it appropriate to recommend a medication review by community pharmacists, patients had at least one DRP (1.3 DRPs per patients). Community pharmacists have to carry out the medication review and monitor the condition of these patients on a regular monthly basis.

In our previous pilot research on another patient population, we found that the interaction risk was the most common cause of DRPs [38]. In this study, the same phenomenon has been found on patients with polypharmacy: the 531 interaction risks detected amounted to 54.0% of all DRPs, which required a deeper understanding of these risks.

Based on the results of the comparative statistical analysis, it can be assumed that the older patients (≥65 years) are more likely to have clinically relevant interaction risks than the younger age group (<65 years). This difference is mainly due to the higher incidence of Grade C interaction risks requiring therapy monitoring, maybe because the number of medications taken is higher in older patients [1], so it has been shown that patients over 65 years require

more special attention during the medication review at a community pharmacy. Monitoring of therapy can be solved within the limits of pharmacist competence, and only in the case of manifested problems it is necessary to consult the GP. This confirmed the need for medication review by pharmacists, as the accumulation of Grade C interaction risks in the elderly can be attributed to the current lack of this service and/or the current lack of a uniform procedure for effective pharmacist-GP collaboration.

The highest proportion of interaction risks were between two prescription drugs (66.7%). There are two possible reasons for a large number of interactions between prescription drugs: the interaction is known to the GP, but the benefit of co-use exceeds the risk carried, as evidenced by the high frequency of interaction risks of Grade C (42.0%), and perhaps the frequency of Grade B (6.4%).

However, based on the occurrence of interaction risks of Grade D and X, it can also be assumed that these problems can also be caused by the overload of GP care and the lack of the medication review role of community pharmacists in Hungary. In all three of these grades of interactions (Grade C, D, and X), it is important to develop an appropriate communication channel between the software of GPs and the software of community pharmacists (e.g. online), because collaboration between GPs and pharmacists is quite complicated by phone or by personal consultation. The prominent role of pharmacists in the detection of interaction risks is also supported by the fact that in 79 cases (14.9%) Grade D and X risk was recorded, which errors would have been hidden without pharmacists' medication reviews, done in community pharmacies.

In this research, in one-third of cases, OTCs or other products (e.g. dietary supplements) caused interaction risk. This result underlines the importance of pharmacists in combating the frequent occurrence of this grade of interaction risks. The solution to these problems is primarily the competence of pharmacists working in community pharmacies, since a significant part of these risks is not visible to the GPs in Hungary, and they do not receive information about them unless the pharmacist indicates them.

Given the high workload of both disciplines (GPs and pharmacists), pharmacists must report the problem to the GP only in the most relevant cases. In our research, 37.1% of the interaction risks identified by pharmacists were clinically irrelevant (Grade A or B risk). The time spent solving these problems places an unnecessary burden on both pharmacists and notified GPs, which in the long term impairs collaboration between the two professionals. In particular, the high detection rate of Grade A interaction risks (30.7%) draws attention to the fact that pharmacists need to be widely and properly educated about the clinical significance of interaction risks, and it is required to standardize interaction risk classifications in drug-dispensing practice in Hungary. Currently, different pharmacy software uses different interaction databases and resources in Hungary during medicine-dispensing, and based on our results, this software does not help to filter out irrelevant interaction risks.

In terms of a comprehensive interpretation of our results, it is clear that certain active substances in the practice of drug dispensing in community pharmacies have a rare clinically relevant risk contrary to the belief of pharmacists (e.g. amlodipine, bisoprolol), whereas special attention is required for certain agents (e.g. acenocoumarol), especially if the medicine is available without a prescription (e.g. diclofenac and acetylsalicylic acid) since the responsibility lies with pharmacists for OTC medicine. Table 6 was compiled based on detailed results to give a practical, conventional device into the hands of pharmacists. Guides such as Table 6 help to recognize the most common clinically significant interaction risks during medicine dispensing in the community pharmacies and also help to memorize the standard high-risk situations in the lack of effective pharmaceutical IT.

In summary, it is important to develop tools for decision support that can be integrated into the pharmacy software, thus, the development of software algorithms to support a

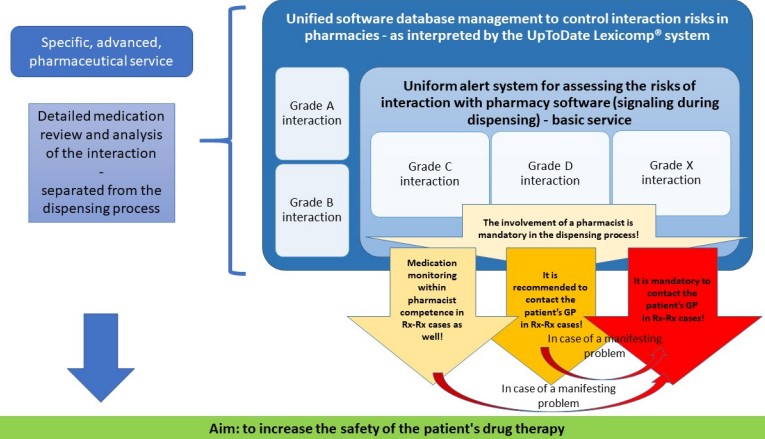

**Fig 7. Proposal for a uniform pharmacy procedure for the management of interaction risks.**

uniform protocol for handling cases of varying severity (e.g. uniform interaction risk classification with the possibility of online notifying to the GPs). A possible IT algorithm to the pharmaceutical protocol is shown in Fig 7. based on a clinical classification of interaction risks according to the UpToDate Lexicomp® database.

Furthermore, in contrast to the high rate of Rx-Rx interaction risks experienced, taking into account the high proportion of prescription drugs in other interaction risks as well (Rx-OTC, RX/OTC-Other), the frequency of pharmacist interventions was low involving GP, even though the most complete and safest solution in these cases, due to the presence of a prescription drug, would be to inform the GP, in addition to other and necessary pharmacist intervention. (This phenomenon is exacerbated by the fact that there is no uniform categorization system in practice regarding the clinical relevance of interaction risks.) Based on our research, it has been found that in addition to (or even instead of) "notification of the GP", a pharmacist can only use "education" (27.4%) as an independent and professional competence.

The need for a uniform pharmaceutical protocol is also supported by the quantitative characterization of the pharmacist interventions recorded in our research to solve the interaction risks, which are far from consistent and not very straightforward. Based on the results of the presented study, pharmacy students need to be purposefully prepared to solve DRPs already at the undergraduate level. In the current Hungarian pharmacy practice, the lack of problem-based educational methodology training may cause that pharmacists consult only about the DPRs to the patients and not to use interventions modifying drug therapy in their own competencies. As a first step, the authors of the study introduced the used project methodology into the curriculum reform of Semmelweis University (e.g. classification and recognition of DRPs according to detected situations) but national improvements are needed in the future.

The development of output requirements for pharmacist training is currently underway in Hungary at the legislative level. In many cases, the rights of the two disciplines (GPs and pharmacists) are not clearly defined in Hungary in solving the interaction risks, so they need to be clearly defined and widely introduced into the pharmacist practice, e.g. what interaction risks should be indicated to the GP in each case (Fig 7). In addition, an effective GP-patient-pharmacist collaboration triangle and the effective medication review are based on adequate communication practice that reduces information asymmetry. Thus, the development of pharmacists' communication skills is also essential in education at both undergraduate and postgraduate levels, especially for community pharmacists because they can help to avoid the DRPs among the population of Hungary.

## Strength and limitation

The project had almost nationwide coverage, including 15 from 20 counties in Hungary. The project included many health care professionals (pharmacists and GPs) and patients with polypharmacy. For the first time in the Hungarian literature, the project assessed the role of community pharmacies in the detection and solution of drug-related problems of patients with polypharmacy. The study examined the interaction risks of patients with polypharmacy entering community pharmacies in terms of their incidence (also relative to all drug-related problems), nature, and clinical severity, and analyzed pharmacist interventions to counter the identified risks. A uniform pharmacy procedure for the management of interaction risks has been proposed by authors.

Selection bias: convenience sample technique was used in the pharmacist, pharmacy, GP, and patient enrolment. The enrolment was also determined by the framework of specialist pharmacist training (eg. number of students, location of accredited pharmacies). No sample size calculation was estimated. Further research is needed to explore the effectiveness of pharmacist interventions to solve interaction risks.

## Conclusions

Based on our results, pharmacists' medication review in community pharmacies is very important for patients with polypharmacy, as a significant number of drug-related problems have been recorded thanks to a complete and detailed overview of the medicines used by patients. Among the problems, the incidence of interaction risks stood out. To solve them effectively, it is essential to develop a unified pharmacy procedure to properly classify the clinical relevance of interaction risks and effectively collaborate with GPs.

## Acknowledgments

The authors would like to thank all the pharmacists, pharmacy technicians, and pharmacies who contributed to the research.

## Author Contributions

**Conceptualization:** András Szilvay, Orsolya Somogyi, Balázs Hankó.

**Data curation:** András Szilvay, Annamária Dobszay.

**Formal analysis:** András Szilvay, Annamária Dobszay, Attiláné Meskó.

**Investigation:** András Szilvay.

**Methodology:** András Szilvay, Orsolya Somogyi.

**Project administration:** András Szilvay.

**Supervision:** Orsolya Somogyi, Romána Zelkó, Balázs Hankó.

**Validation:** Orsolya Somogyi.

**Visualization:** András Szilvay, Orsolya Somogyi.

**Writing – original draft:** András Szilvay.

**Writing – review & editing:** Orsolya Somogyi, Romána Zelkó.

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
