## [Decision Letter · Decision Letter 0]

8 Apr 2021

PONE-D-21-08091

Analysis of interaction risks of patients with polypharmacy and the pharmacist interventions performed to solve them – a multicenter descriptive study according to medication reviews in Hungarian community pharmacies

PLOS ONE

Dear Dr. Szilvay,

Thank you for submitting your manuscript to PLOS ONE. After careful consideration, we feel that it has merit but does not fully meet PLOS ONE’s publication criteria as it currently stands. Therefore, we invite you to submit a revised version of the manuscript that addresses the points raised during the review process.

We look forward to receiving your revised manuscript.

Kind regards,

Vijayaprakash Suppiah, PhD

Academic Editor

PLOS ONE

Journal Requirements:

We note that you have stated that you will provide repository information for your data at acceptance. Should your manuscript be accepted for publication, we will hold it until you provide the relevant accession numbers or DOIs necessary to access your data. If you wish to make changes to your Data Availability statement, please describe these changes in your cover letter and we will update your Data Availability statement to reflect the information you provide.

Reviewers' comments:

Reviewer's Responses to Questions

**Comments to the Author**

1. Is the manuscript technically sound, and do the data support the conclusions?

Reviewer #1: Yes

Reviewer #2: Yes

Reviewer #3: Partly

Reviewer #4: Partly

Reviewer #5: Yes

2. Has the statistical analysis been performed appropriately and rigorously? 

Reviewer #1: Yes

Reviewer #2: Yes

Reviewer #3: N/A

Reviewer #4: I Don't Know

Reviewer #5: Yes

3. Have the authors made all data underlying the findings in their manuscript fully available?

Reviewer #1: Yes

Reviewer #2: No

Reviewer #3: Yes

Reviewer #4: No

Reviewer #5: Yes

4. Is the manuscript presented in an intelligible fashion and written in standard English?

Reviewer #1: Yes

Reviewer #2: Yes

Reviewer #3: No

Reviewer #4: Yes

Reviewer #5: Yes

5. Review Comments to the Author

Reviewer #1: The manuscript is well written and highlights the potential drug interaction related issues in polypharmacy. The role of Pharmacists in managing the risks associated with drug interactions is well established.

Reviewer #2: The study on drug-related problems, especially the interaction risks of patients with polypharmacy entering community pharmacies in terms of their nature, and clinical severity and the pharmacist interventions performed to solve them. It appears to be a good study because of the increasing number of chronic conditions and the need to use a number of medications, which increases the risk of drug-related problems and reduces medication adherence, further halting the overall outcome. The authors of this study correctly identified the problems and devised solutions to address them. Because HCPs and patients may lack knowledge or awareness about DRPs, they provide education before beginning the project.

However, there are issues need to be addressed before it can be considered for publication.

Abstract

Kindly make structure abstract, objective, methodology, results and conclusion

Introduction

It will be good to talk about the need of the study? What is the rational for the study? From previous literature available from Hungry or other countries, if available such data. Please highlighted in the introduction part.

Methods

Excellent methodology and literature review however the authors miss about the researcher, how they trained or one of the author/researcher involve in data collection and analysis?

There is a need to describe the basic country geography, such as how many counties/cities/towns there are, and so on.

Which methodology was used to select participants to represent the study across the country?

Is a sample size calculation performed prior to the actual study? What is the starting point for including 78 pharmacists, 98 general practitioners, and 755 patients?

Please explain in methodology section

Ethics approval

Why didn't the researchers apply to the Human Research Ethics Committee (HREC)?

All research projects involving human participants, including the use of non-face-to-face data collection methods such as online surveys and postal surveys, require HREC approval.

I understand that ethical approval is not required in their country, as researchers stated.

I believe they should still apply for HREC; the HREC committee may grant them exemption based on the project is negligible risk.

General comments

kindly check for typographical error

Please follow uniform pattern for reference style, refer journal guidelines

The strength and the limitations of the study? Make it as separate heading

I think if the corrections are made the paper can be accepted for publication in PloS ONE.

Reviewer #3: The authors state that “All dara are available from https://docs.google.com/spreadsheets/d/1xNOYnVJNZx-

h3CyDbyvIgZQ_yCSkoKl7/edit#gid=454092672.” However, trying to access the data we found that “You need access”. Therefore, the authors should provide free access to the data, otherwise currently they are not available.

We also tend to agree that for such cases no ethical approval should be needed, but the majority of authors and countries tend to ask for ethical approval also in such cases. Particularly the oral informed consent cannot be proven in the absence of a written document.

Line 19: the phrase “enhance their practical effectiveness” is unclear in its object and should be rephrase to clarify to what “practical effectiveness” refers to (obviously not to “identified risks”, but now the phrasing is slightly confusing).

Lines 73-75: reduction of effectiveness is only one sense of potential DDI, the other the additive effect, also with increased morbidity and mortality (but through increased effectiveness).

Lines 90-91: it is not clear if the study is affected by a selection bias, because no information is provided on the response rate of the patients (what proportion of patients entering a pharmacy were willing to take part in the project activities? Was there any significant difference between patients accepting participation and those refusing participation?)

Table 1: please define/explain the meaning of “Non-quantitative ineffectiveness” and “non-quantitative safety problems”. It seems to us that something like “dosage-related ineffectiveness” and “ineffectiveness related to other causes than dosage” would be clearer (and equivalent phrasing for safety problems).

Table 4, first row: “In the case of interaction risks, e.g., pharmacokinetic interactions, this may mean a change in the time of using the medication.” This should be phrased in a clearer manner, “change in the time” may be used in the sense of dosing interval (tau) (change in the time between two successive doses) or in the sense of changing the moment of the day when the product is administered etc.

For “Warning the GP “, replacing with “Notification of the GP” would be more polite.

Line 278: the meaning of the parenthesis (“(perhaps Grade B)” is not clear.

The paper includes no “Limitations” section, although one would be strongly needed. For instance, the sample of DRPs and DDIs seem to be part of more or less a convenience sample, and therefore it is not clear to what extent the findings are really representative at the national level in Hungary (despite the claimed “nationwide coverage”).

Figure 2 – considering the use of the English language, the comma should be replaced by a point (e.g. 60.0% instead of 60,0%).

Figure 3 – it is widely known that human eye and brain are incapable of comparing circle angles and slices (see e.g. https://www.bernardmarr.com/default.asp?contentID=1779 or https://scc.ms.unimelb.edu.au/resources-list/data-visualisation-and-exploration/no_pie-charts ). A bar chart is highly preferable.

Figure 4 – there must be a legend accompanying this figure (to explain the meaning of each color).

The English language needs moderate changes and many sentences are difficult to follow, they could be rephrased for easier reading.

Reviewer #4: 1. Please correct the spelling of “data” in data availability statement

2. Abstract of the current study requires particular attention. Authors should concisely describe the methodology, number of participants, primary outcomes, major findings. Moreover, conclusion must be align with the findings with specific remarks

3. This manuscript needs considerable efforts to improve the scientific writing. Though authors provided the sentences with good English, but still essence of scientific writing is missing in the draft.

4. I will suggest authors to make heading of “Operational definitions” and put all the terminologies under this heading

5. The methodology section requires appropriate demarcation i.e. study design, sample size estimation, study duration, study population with inclusion and exclusion criterion, data collection, study variables. There is no detail of statistical analysis in the draft.

6. Please provide the English translation of reference 44; I will suggest authors to explain bit about the DRPs classification as provided reference is not in English.

7. Please provide the operational definition of polypharmacy used in the current study

8. The presentation of figure 3 is confusing, can authors use another format to present the result e.g. stacked column or stacked bar

9. Can authors provide relationship analysis between demographic variables with severity of DRPs. I am convinced that classification of DRPs based on severity or necessity for intervention is of great value. Pharmacists can ascertain the particular class of patients vulnerable to such DRPs and could avoid such instances in the future.

10. In 755, 984 DRPs were reported, indicating that all the patients visiting the pharmacies experienced DRPs. It is not clear that authors took data of all patients admitting to pharmacies during the specified time period or they took only those patients who had DRPs.

11. During the 6 months duration of the study, only 755 patients attended the pharmacies located in 35 settlements?

12. 15 out of 20 countries???? This statement is not clear

13. Please provide the details of covered area so generalizability of the findings can be assessed.

14. Were all these pharmacies included in the current study belong to same owner or cooperation so they were randomly selected?

15. The pharmacy selection criteria are needed to be explained as authors claimed that current study has nationwide coverage.

16. How the sampling was done in the current study? Who collected the data? How DRPs identified (by pharmacist or by software)?

17. Authors need to discuss about the possible bias in the current study

18. Conclusion of the study needs attention, it should be specific to the findings and with future suggestions

19. 6. The incidence of pharmacist interventions (Table 4) to solve interaction risks, in order of …. There is no need to indicate table number in the figure legend

20. Since warning of GP and education was most common intervention in the current study, authors need to underscore the reason behind that as very less data is available showing the role of pharmacist as dosage modifier or drug changer? Do pharmacists consider therapy modification bit risky and challenging as compared to other interventions?

Reviewer #5: I read with pleasure and curiosity your manuscript titled "Analysis of interaction risks of patients with polypharmacy and the pharmacist interventions performed to solve them – a multicenter descriptive study according to medication reviews in Hungarian community pharmacies."

I feel this manuscript provides the international readers with relevant information on the community pharmacists' practices in Hungary. Pharmaceutical care planning, medication [utilisation] reviews and clinical roles of the community phamracists are everchanging and evolving roles that underpin the effectiveness of the current pharmacy education strategies.

Two broad comments:

1- Please make sure you use the international recognised language of pharmaceutical care. For example, you have used the verb "warned" in your manuscript. This is not an appropriate verb when it comes to pharmaceutical care planning, communication with the prescribers and pharmacists' intervention re ADRs. We wont warn our colleagues who prescribe the medications. We communicate with them and explain the potential issues with their prescribing and provide them with solutions to address the potential issue.

I highly recommend you to thoroughly read your manuscript and carefully address this issue of the language.

2- Please refer to the standard books/resources of pharmaceutical care. medication reviewe, etc. such as Michael E Winter book on pharmaceutical care, Linda Strand's body of work on pharmaceutical care and Steve Hudson's iconinc work on pharmaceutical care. Methinks referring to these internationally recognised resources, would help in using the classifications/terminologies that are known to a wider range of readers.

For example the classification of the DRPs (table 1) was totally new to me. I tried to match this classification with the Michael E Winter's seminal work on the classification of DRPs; and I failed to do so. I had never heard of reference of 44 that is apparently the resource that you have derived table 1 from.

Introduction:

1- A background information on the Hungarian Pharmacy Education and the current Pharmacy Practice is missing.

I suggest instead of writing about the types of DDIs (that should be a known knowledge), you add the information that would help you better explain and discuss your findings.

2- The introduction references need updating. For example, the polypharmacy information (line 42) is 11 years old!

Methods:

Line 96: It was not clear to me how a one-day course would help the pharmacists familiarise themselves with the process and the requirement of the research. What were the contents of the one-day course? Who developed them? Were these contents validated? Was the classification system for the DRPs the one that is widely available to all of the Hungarian community pharmacists?

- Were the 98 GPs also given the one-day course? If not, what was the extend of their involvement in this research?

I am more interested to know how and if the Hungarian Pharmacy Education prepares the pharmacists to get involved in medication review, pharmaceutical care planning processes and to identify potential DDIs rather than reading about the definitions of interventions, etc. I dont find Table 4 , for instance, of a perticular benefit to the readers.

1- How did you ensure that the patients who were on 5 or medications actually took all of them? How concordance was evaluated, if it was at all?

2- Lines 127 & 128 : What do you mean by "pharmacists recorded the active substances involved..."?

3- Table 2: I suppose you mean "collaboration" rather than "cooperation"?

4- Table 3: Did you double check the grouping of the interactions from a clinical risk perspective with other resources such as Stockly's Drug Interactions?

5- Table 5: Can you double check the accuracy of the information re "The average number of products +- S.D?

6- Does product mean medicine?

Results:

1- What are the definitions of "Quantitative Safety problem" and " Non-quantitative safety problem? Are these standard terminologies? if yes, please define them, first.

2- Product, Medicine, drugs, active ingredients, active substances are the terms that need to be clearly defined and consistantly used throughout the manuscript. In its current form, the terms are confusing.

3- Overall, you have focused on "What" you found! For an international readers, the "Hows" of addressing "What" was found is more interesting/educational. In the same light, you have only written 4 sentences on the analysis of pharmacist intervetions to slove interaction risks?!

4- I wishfully think that you could also run some simple regression analysis to investigate some associations between patients characteristics and the types/numbers/severity of DDI or DRPs. In its current form, all of the statistical analyses have been purely descriptive.

Discussion:

Line 272: 531 interaction risks per patient? I take this figure with a pinch of salt! This is too high to me and if it's true it needs to be discussed in details.

Line 280: It feels that you are pointing finger at the lack of clear communications between patients and physicians, only!

I don't think this would be of any help to anyone. Pharmacists in the communities are meant to fill in that gap. So, if the gap is not filled, then the community pharmacists do need to be included in the discussion.

Line 287: Instead of 1/3 you could write it in full i.e., one third...

Line 298: To reiterate my ealier point on the need to discuss the current pharmacy education and practices in Hungary in much more details.

Line 308: Table 6 in the discussion? What does it add to the discussion that is relevant to the research findings?

- To appreciate Figure 7 better, one would need to know about the current practices and how Figure 7 is better if at all.

6. PLOS authors have the option to publish the peer review history of their article (what does this mean?). If published, this will include your full peer review and any attached files.

Reviewer #1: No

Reviewer #2: **Yes: **Akram Ahmad

Faculty of Medicine and Health,

the University of Sydney, Australia

Reviewer #3: No

Reviewer #4: **Yes: **Abdullah Salah Alanazi

Reviewer #5: **Yes: **Keivan Ahmadi

---

## [Author Response · Author response to Decision Letter 0]

28 May 2021

Answers to the Reviewers' comments:

Thank you very much for your valuable comments, We tried to modify the manuscript accordingly. 

Reviewer #1

1. The manuscript is well written and highlights the potential drug interaction related issues in polypharmacy. The role of Pharmacists in managing the risks associated with drug interactions is well established.

a. Thank you for your review.

Reviewer #2

Abstract

1. Kindly make structure abstract,objective, methodology, results and conclusio

o Abstract has been reworded.

Introduction

2. It will be good to talk about the need of the study? What is the rational for the study? From previous literature available from Hungry or other countries, if available such data. Please highlighted in the introduction part.

o Line 92: „In Hungary, community pharmacists have a statutory task of detecting clinically significant interaction risks in community pharmacies within the framework of pharmaceutical care [36]. A study looked at the incidence of 39 potentially dangerous interactions: these each occurred in 0-335.89 / 100,000 prescriptions per year. [37] Previous research showed, that the interaction risk was the most common cause of DRPs detected in Hungarian community pharmacies, [38] but at present, little real-life information is known on the clinical relevance of interaction risks, and pharmacist interventions to prevent them.”

Methods

3. Excellent methodology and literature review however the authors miss about the researcher, how they trained or one of the author/researcher involve in data collection and analysis?

o Line 124: „At the beginning of the project, pharmacists received a one-day course at Semmelweis University, during which participating pharmacists were introduced to the detailed goals, implementation steps, and professional content to be used.”

o Line 160: „From the data collected by pharmacists, the authors determined the amount of DRPs and interaction risks per capita and the percentage of each DRP category and underlying cause in proportion to the total DRP.”

4. There is a need to describe the basic country geography, such as how many counties/cities/towns there are, and so on.

o The curiosity of the reviewer is understandable.

o Line 221: „The project covered 15 from 20 counties (including Budapest, the capital of Hungary)[…]”

5. Which methodology was used to select participants to represent the study across the country?

o Line 110: “The research was carried out with the participation of graduated pharmacists (they did not receive monetary compensation). No randomization was used in the selection of participating pharmacists, in Hungarian community pharmacies serving as their workplace, which were accredited pharmacies at Semmelweis University.”

o Line 114: „The enrollment of the patients was made by pharmacists in community pharmacies using convenience sample technique.”

o Line 123: “The research was carried out in the framework of the training of specialist pharmacists at Semmelweis University.”

6. Is a sample size calculation performed prior to the actual study? What is the starting point for including 78 pharmacists, 98 general practitioners, and 755 patients?

Please explain in methodology section.

o We did not use sample size calculation.

o Line 110: „The research was carried out with the participation of graduated pharmacists (they did not receive monetary compensation). No randomization was used in the selection of participating pharmacists, in Hungarian community pharmacies serving as their workplace, which were accredited pharmacies at Semmelweis University.

The enrollment of the patients was made by pharmacists in community pharmacies using convenience sample technique. Every pharmacist had to invite nearly 10 patients to participate in the project. The survey involved volunteers over the age of 18 with polypharmacy (using 5 or more drugs continuously [4, 5]) who got their medications monthly. Patients were invited to participate based on whether a detailed medication review by pharmacists was warranted based on the professional opinion of the patients' general practitioners (GPs).

For the project to be successful, each pharmacist had to try to include at least 1 GP in the research whose patients receive their medicines at that pharmacy.”

Ethics approval

7. Why didn't the researchers apply to the Human Research Ethics Committee (HREC)?

All research projects involving human participants, including the use of non-face-to-face data collection methods such as online surveys and postal surveys, require HREC approval.

I understand that ethical approval is not required in their country, as researchers stated.

I believe they should still apply for HREC; the HREC committee may grant them exemption based on the project is negligible risk.

o In Hungary according to the regulations, pharmacies did not need to be individually ethically licensed, because the service complies with statutory regulations, and pharmacies are legally entitled to perform such activities. However, the research has been accepted by Semmelweis Iniversity Regional and Institutional Committee of Science and Research Ethics (SE RKEB: 110/2021).

o Line 205: „However, the research has been accepted by Semmelweis University Regional and Institutional Committee of Science and Research Ethics (SE RKEB: 110/2021).”

General comments

8. kindly check for typographical error

o The manuscript has been checked for typographical errors.

9. Please follow uniform pattern for reference style, refer journal guidelines

o Reference style has been checked.

10. The strength and the limitations of the study? Make it as separate heading.

o The „strength and limitation” section has been inserted in Line 423.

11. I think if the corrections are made the paper can be accepted for publication in PloS ONE.

o Thank you for your comments.

Reviewer #3

1. The authors state that “All dara are available from https://docs.google.com/spreadsheets/d/1xNOYnVJNZx-

h3CyDbyvIgZQ_yCSkoKl7/edit#gid=454092672.” However, trying to access the data we found that “You need access”. Therefore, the authors should provide free access to the data, otherwise currently they are not available.

a. All data are available at: https://drive.google.com/file/d/1xNOYnVJNZx-h3CyDbyvIgZQ_yCSkoKl7/view?usp=sharing

2. We also tend to agree that for such cases no ethical approval should be needed, but the majority of authors and countries tend to ask for ethical approval also in such cases. Particularly the oral informed consent cannot be proven in the absence of a written document.

a. In Hungary according to the regulations, pharmacies did not need to be individually ethically licensed, because the service complies with statutory regulations, and pharmacies are legally entitled to perform such activities. However, the research has been accepted by Semmelweis Iniversity Regional and Institutional Committee of Science and Research Ethics (SE RKEB: 110/2021).

b. Line 205: „However, the research has been accepted by Semmelweis University Regional and Institutional Committee of Science and Research Ethics (SE RKEB: 110/2021).”

3. Line 19: the phrase “enhance their practical effectiveness” is unclear in its object and should be rephrase to clarify to what “practical effectiveness” refers to (obviously not to “identified risks”, but now the phrasing is slightly confusing).

a. Abstract has been reworded.

b. Line 100: “Also, the objective was to analyze and enhance the practical effectiveness of pharmacist interventions to counter the identified risks and prepare a procedure for the uniform handling of drug interactions.”

4. Lines 73-75: reduction of effectiveness is only one sense of potential DDI, the other the additive effect, also with increased morbidity and mortality (but through increased effectiveness).

a. Line 73: „DDIs carry a serious health risk: by reducing (or enhance by an additive effect) the effectiveness of therapy, they increase morbidity and mortality [30] and increase the risk of hospital admission, which also places a financial burden on the healthcare system. [31, 32]”

5. Lines 90-91: it is not clear if the study is affected by a selection bias, because no information is provided on the response rate of the patients (what proportion of patients entering a pharmacy were willing to take part in the project activities? Was there any significant difference between patients accepting participation and those refusing participation?)

a. Line 432: „Selection bias: convenience sample technique was used in the pharmacist, pharmacy, GP, and patient enrolment. The enrolment was also determined by the framework of specialist pharmacist training (eg. number of students, location of accredited pharmacies). No sample size calculation was estimated.”

6. Table 1: please define/explain the meaning of “Non-quantitative ineffectiveness” and “non-quantitative safety problems”. It seems to us that something like “dosage-related ineffectiveness” and “ineffectiveness related to other causes than dosage” would be clearer (and equivalent phrasing for safety problems).

a. „Non-quantitative ineffectiveness” and „non-quantitative safety problems” are the terms used by the developers of the DRP classification system (reference 41). Non-quantitative ineffectiveness/safety problems mean drug-related ineffectiveness/safety problems, while quantitative ineffectiveness/safety problems mean that the DRP depends on the magnitude of an effect.

b. Line 152: „*Non-quantitative DRP: the DRP is drug-related, it does not depend on the magnitude of an effect. **Quantitative DRP: The DRP depends on the magnitude of an effect.”

7. Table 4, first row: “In the case of interaction risks, e.g., pharmacokinetic interactions, this may mean a change in the time of using the medication.” This should be phrased in a clearer manner, “change in thetime” may be used in the sense of dosing interval (tau) (change in the time between two successive doses) or in the sense of changing the moment of the day when the product is administered etc.

a. The sentence has been rephrased: „DRP has been resolved by a change in the dosage regimen of a given medicine recommended by the pharmacist. In the case of interaction risks, e.g., pharmacokinetic interactions, this may mean a change in the moment of the day when the drug is administrated.

8. For “Warning the GP “, replacing with “Notification of the GP” would be more polite.

a. Warning the GP has been replaced with „Notification of the GP”.

9. Line 278: the meaning of the parenthesis (“(perhaps Grade B)” is not clear.

a. The sentence has been rephrased in Line 352: „…the interaction is known to the GP, but the benefit of co-use exceeds the risk carried, as evidenced by the high frequency of interaction risks of Grade C (42.0%), and perhaps the frequency of Grade B (6.4%)”

10. The paper includes no “Limitations” section, although one would be strongly needed. For instance, the sample of DRPs and DDIs seem to be part of more or less a convenience sample, and therefore it is not clear to what extent the findings are really representative at the national level in Hungary (despite the claimed “nationwide coverage”).

a. The „strength and limitation” section has been inserted in Line 423.

11. Figure 2 – considering the use of the English language, the comma should be replaced by a point (e.g. 60.0% instead of 60,0%).

a. Figures 2, 3, 4, 5, and 6 have been corrected according to the review.

12. Figure 3 – it is widely known that human eye and brainare incapable of comparing circle angles and slices (see e.g. https://www.bernardmarr.com/default.asp?contentID=1779 or https://scc.ms.unimelb.edu.au/resources-list/data-visualisation-and-exploration/no_pie-charts ). A bar chart is highly preferable.

a. Figure 3 has been replaced.

13. Figure 4 – there must be a legend accompanying this figure (to explain the meaning of each color).

a. Line 278: „Fig 4. Distribution of interactions by UpToDate Lexiomp® risk classification grades. [43] A: no known interaction, B: no action needed, C: monitor therapy, D: consider therapy modification, X: avoid combination; n.a.: not available; n=531.”

14. The English language needs moderate changes and many sentences are difficult to follow, they could be rephrased for easier reading.

a. The language of the manuscript has been corrected.

Reviewer #4:

1. Please correct the spelling of “data” in data availability statement

a. The statement has been corrected.

2. Abstract of the current study requires particular attention. Authors should concisely describe the methodology, number of participants, primary outcomes, major findings. Moreover, conclusion must be align with the findings with specific remarks.

a. Abstract has been reworded.

3. This manuscript needs considerable efforts to improve the scientific writing. Though authors provided the sentences withgood English, but still essence of scientific writing is missing in the draft.

a. The manuscript was corrected as requested.

4. I will suggest authors to make heading of “Operational definitions” and put all the terminologies under this heading.

a. Thanks for the comment. Collecting definitions separately is a good idea, however, in our opinion, describing the definitions at the first mention provides better comprehensibility when reading the manuscript. Following the review, we paid close attention to the explanatory description of the definitions in the manuscript.

5. The methodology section requires appropiate demarcation i.e. study design, sample size estimation, study duration, study population with inclusion and exclusion criterion, data collection, study variables. There is no detail of statistical analysis in the draft.

a. The methodology section has been rearranged.

b. Methods, Results, and Discussion section have been completed with statistical analysis requested.

c. „Line 182: „The frequency of interaction risks considered to be clinically relevant was compared by two demographic aspects (gender, age). We examined the incidence of Grade C, D, or X interaction risks separately in men and women, and in patients under or more than 65 years. In addition, the combined incidence of Grade C, D, and X interaction risks was also examined for both aspects (gender, age).”

d. Line 199: “Based on the descriptive statistics, in the case of Grade C, D, or X interaction risks (separately, and together), the relationship between "one or more interactions" versus "no interactions" by gender and age was examined by the chi-square test. The significance level was 5%.”

e. Line 305: „In comparative studies by gender, the incidence of Grade C, D, or X interaction risks was examined separately and then aggregated (C + D + X). No significant difference was found between men and women in either case (p>0.05). Examining the age groups, we found that there was a more frequent Grade C interaction risk in the age group 65 years or older, with a significant difference (p=0.05), while no significant difference was found for Grade D or X interaction risks. Looking at the combined incidence of Grade C, D, and X interaction risks, it can be assumed in professional practice that the older (≥ 65 years) age group is more likely to have clinically relevant interaction risks than the younger age group (p=0.076, close to the significance limit).”

f. Line 341: “Based on the results of the comparative statistical analysis, it can be assumed that the older patients (≥65 years) are more likely to have clinically relevant interaction risks than the younger age group (<65 years). This difference is mainly due to the higher incidence of Grade C interaction risks requiring therapy monitoring, maybe because the number of medications taken is higher in older patients [1], so it has been shown that patients over 65 years require more special attention during the medication review at a community pharmacy. Monitoring of therapy can be solved within the limits of pharmacist competence, and only in the case of manifested problems it is necessary to consult the GP. This confirmed the need for medication review by pharmacists, as the accumulation of Grade C interaction risks in the elderly can be attributed to the current lack of this service and/or the current lack of a uniform procedure for effective pharmacist-GP collaboration.“

6. Please provide the English translation of reference 44; I will suggest authors to explain bit about the DRPs classification as provided reference is not in English.

a. The article (reference 41) is available in English from: http://citeseerx.ist.psu.edu/viewdoc/download?doi=10.1.1.409.4718&rep=rep1&type=pdf

b. Line 147: „The classification system used was chosen and used in the research on the basis of previous successful Hungarian projects. [38, 40]”

7. Please provide the operational definition of polypharmacy used in the current study.

a. Line 50: „According to the most widely used definition (as in this manuscript): polypharmacy means the continuous concomitant use of 5 or more drugs. [4, 5]”

8. The presentation of figure 3 is confusing, can authors use another format to present the result e.g. stacked column or stacked bar.

a. Figure 3 has been replaced

9. Can authors provide relationship analysis between demographic variables with severity of DRPs. I am convinced that classification of DRPs based on severity or necessity for intervention is of great value. Pharmacists can ascertain the particular class of patients vulnerable to such DRPs and could avoid such instances in the future.

a. Methods, Results and Discussion section have been completed with statistcal analysis requested.

b. Line 182: „The frequency of interaction risks considered to be clinically relevant was compared by two demographic aspects (gender, age). We examined the incidence of Grade C, D, or X interaction risks separately in men and women, and in patients under or more than 65 years. In addition, the combined incidence of Grade C, D, and X interaction risks was also examined for both aspects (gender, age).”

c. Line 199: “Based on the descriptive statistics, in the case of Grade C, D, or X interaction risks (separately, and together), the relationship between "one or more interactions" versus "no interactions" by gender and age was examined by the chi-square test. The significance level was 5%.”

d. Line 305: „In comparative studies by gender, the incidence of Grade C, D, or X interaction risks was examined separately and then aggregated (C + D + X). No significant difference was found between men and women in either case (p>0.05). Examining the age groups, we found that there was a more frequent Grade C interaction risk in the age group 65 years or older, with a significant difference (p=0.05), while no significant difference was found for Grade D or X interaction risks. Looking at the combined incidence of Grade C, D, and X interaction risks, it can be assumed in professional practice that the older (≥ 65 years) age group is more likely to have clinically relevant interaction risks than the younger age group (p=0.076, close to the significance limit).”

e. Line 341: “Based on the results of the comparative statistical analysis, it can be assumed that the older patients (≥65 years) are more likely to have clinically relevant interaction risks than the younger age group (<65 years). This difference is mainly due to the higher incidence of Grade C interaction risks requiring therapy monitoring, maybe because the number of medications taken is higher in older patients [1], so it has been shown that patients over 65 years require more special attention during the medication review at a community pharmacy. Monitoring of therapy can be solved within the limits of pharmacist competence, and only in the case of manifested problems it is necessary to consult the GP. This confirmed the need for medication review by pharmacists, as the accumulation of Grade C interaction risks in the elderly can be attributed to the current lack of this service and/or the current lack of a uniform procedure for effective pharmacist-GP collaboration. “

10. In 755, 984 DRPs were reported, indicating that all the patients visiting the pharmacies experienced DRPs. It is not clear that authors took data of all patients admitting to pharmacies during the specified time period or they took only those patients who had DRPs.

a. We took data of all patients who agreed to participate in the project during the specified period. There were patients with no DRPs in the project.

b. Line 114: „The enrollment of the patients was made by pharmacists in community pharmacies using convenience sample technique. Every pharmacist had to invite nearly 10 patients to participate in the project. The survey involved volunteers over the age of 18 with polypharmacy (using 5 or more drugs continuously [4, 5]) who got their medications monthly. Patients were invited to participate based on whether a detailed medication review by pharmacists was warranted based on the professional opinion of the patients' general practitioners (GPs).”

11. During the 6 months duration of the study, only 755 patients attended the pharmacies located in 35 settlements?

a. 755 patients agreed to participate in the project in 35 settlements.

b. Line 114: „The enrollment of the patients was made by pharmacists in community pharmacies using convenience sample technique. Every pharmacist had to invite nearly 10 patients to participate in the project.”

12. 15 out of 20 countries???? This statement is not clear

a. The sentence was misread. Hungary has 20 counTIES, including the capital.

b. Line 221: „The project covered 15 from 20 counties (including Budapest, the capital of Hungary) [...]”

13. Please provide the details of covered area so generalizability of the findings can be assessed.

a. Line 221: “The project covered 15 from 20 counties (including Budapest, the capital of Hungary), 35 settlements, so the survey can be considered to have almost nationwide coverage.”

14. Were all these pharmacies included in the current study belong to same owner or cooperation so they were randomly selected?

a. In Hungary, most pharmacies are privately owned by pharmacists. 

b. Line 110: „The research was carried out with the participation of graduated pharmacists (they did not receive monetary compensation). No randomization was used in the selection of participating pharmacists, in Hungarian community pharmacies serving as their workplace, which were accredited pharmacies at Semmelweis University.”

15. The pharmacy selection criteria areneeded to be explained as authors claimed that current study has nationwide coverage.

a. The sentence quoted has been rewritten.

b. Line 221: “The project covered 15 from 20 counties (including Budapest, the capital of Hungary), 35 settlements, so the survey can be considered to have almost nationwide coverage.”

c. Line 110: „The research was carried out with the participation of graduated pharmacists (they did not receive monetary compensation). No randomization was used in the selection of participating pharmacists, in Hungarian community pharmacies serving as their workplace, which were accredited pharmacies at Semmelweis University.”

16. How the sampling was done in the current study? 

a. Convenience sample technique was used in the pharmacist, pharmacy, GP, and patient enrolment. 

17. Who collected the data? 

a. Data were collected by participating pharmacists, at their workplace, then they forwarded them to the authors anonymously for further analysis.

b. Line 160: „From the data collected by pharmacists, the authors determined the amount of DRPs and interaction risks per capita and the percentage of each DRP category and underlying cause in proportion to the total DRP.”

18. How DRPs identified (by pharmacist or by software)?

a. DRPs were identified by a pharmacist according to the methodology described in the one-day course.

b. Line 140: „In the event of a DRP detected by the pharmacist, a pharmacist intervention was carried out […]”

19. Authors need to discuss about the possible bias in the current study

a. The „strength and limitation” section has been inserted in Line 423, according to other reviewers' opinions.

20. Conclusion of the study needs attention, it should be specific to the findings and with future suggestions

a. The discussion section was completely rewritten.

21. The incidence of pharmacist interventions (Table 4) to solve interaction risks, in order of …. There is no need to indicate table number in the figure legend

a. The figure legend has been corrected.

b. Line 325: „Fig 6. The incidence of pharmacist interventions to solve interaction risks, in order of frequency.”

22. Since warning of GP and education was most common intervention in the current study, authors need to underscore the reason behind that as very less data is available showing the role of pharmacist as dosage modifier or drug changer? Do pharmacists consider therapy modification bit risky and challenging as compared to other interventions?

a. The discussion section was completely rewritten.

b. Line 406: “Based on the results of the presented study, pharmacy students need to be purposefully prepared to solve DRPs already at the undergraduate level. In the current Hungarian pharmacy practice, the lack of problem-based educational methodology training may cause that pharmacists consult only about the DPRs to the patients and not to use interventions modifying drug therapy in their own competencies. As a first step, the authors of the study introduced the used project methodology into the curriculum reform of Semmelweis University (e.g. classification and recognition of DRPs according to detected situations) but national improvements are needed in the future.”

Reviewer #5:

Two broad comments:

1. Please make sure you use the international recognised language of pharmaceutical care. For example, you have used the verb "warned" in your manuscript. This is not an appropriate verb when it comes to pharmaceutical care planning, communication with the prescribers and pharmacists'intervention re ADRs. We wont warn our colleagues who prescribe the medications. We communicate with them and explain the potential issues with their prescribing and provide them with solutions to address the potential issue.

I highly recommend you to thoroughly read your manuscript and carefully address this issue of the language.

a. The manuscript was corrected as requested.

2. Please refer to the standard books/resources of pharmaceutical care. medication reviewe, etc. such as Michael E Winter book on pharmaceutical care, Linda Strand's body of work on pharmaceutical care and Steve Hudson's iconinc work on pharmaceutical care. Methinks referring to these internationally recognised resources, would help in using the classifications/terminologies that are known to a wider range of readers.

For example the classification of the DRPs (table 1) was totally new to me. I tried to match this classification with the Michael E Winter's seminal work on the classification of DRPs; and I failed todo so. I had never heard of reference of 44 that is apparently the resource that you have derived table 1 from.

a. Thanks for your recommendation, the manuscript has been corrected accordingly.

b. Line 147: „The classification system used was chosen and used in the research on the basis of previous successful Hungarian projects. [38, 40]” 

c. Reference 41 is available in English from: http://citeseerx.ist.psu.edu/viewdoc/download?doi=10.1.1.409.4718&rep=rep1&type=pdf

Introduction:

3. A background information on the Hungarian Pharmacy Education and the current Pharmacy Practice is missing. I suggest instead of writing about the types of DDIs (that should be a known knowledge), you add the information that would help you better explain and discuss your findings.

a. Line 81: „In Hungary, pharmacists receive a degree after five years at university. In the first two years, basic science knowledge (mathematics, chemistry, biology, botany, etc.) is taught to students, while in the second and third years, they study basic medical knowledge (cell biology, biochemistry, physiology, etc.). This knowledge is the basis of pharmaceutical subjects in the 4th and 5th years, including the study of pharmaceutical care as a separate subject for one semester, during which they get acquainted with the most important therapeutic situations in community pharmacies (antibiotic use, bandages, asthma care, diabetes care, etc.), mainly with theoretical education. 

The health care institutions that cover Hungary most evenly are the community pharmacies operating as part of the primary care. The majority of patients visit pharmacies for two reasons: 1) to get a drug prescribed by a general practitioner or a specialist; 2) to seek advice on relieving their mild symptoms. During a consultation, pharmacists dispense the prescribed drug or recommend an over-the-counter (OTC) medication for the patient's symptoms.”

4. The introduction references need updating. For example, the polypharmacy information (line 42) is 11 years old!

a. The introduction references have been updated.

Methods:

5. Line 96: It was not clear to me how a one-day course would help the pharmacists familiarise themselves with the process and the requirement of the research. What were the contents of the one-day course? Who developed them? Were these contents validated? Was the classification system for the DRPs the one that is widely available to all of the Hungarian community pharmacists?

a. Line 124: “At the beginning of the project, pharmacists received a one-day course at Semmelweis University, during which participating pharmacists were introduced to the detailed goals, implementation steps, and professional content to be used. To implement the project, a statutory professional guideline, [39], as well as the methodological bases of „Metabolic Syndrome Pharmacological Care Program 2.0.” (see the classification of drug-related problems) [45], were used by pharmacists, which professional materials were available to all pharmacists before the presented study. The tools to ensure the practical use of these materials (e.g. tables, questionnaires) and the procedure for documentation have been developed by the authors of the manuscript based on the experience of previous pilot projects [38].”

6. Were the 98 GPs also giventhe one-day course? If not, what was the extend of their involvement in this research?

a. Line 133: “The GPs involved received a written summary of the project implementation steps, and the content of the one-day training was available online with the help of the cooperating pharmacists. In addition to assisting pharmacists in involving patients, GPs provided help in resolving DRPs if they were approached by participating pharmacists with the problem.”

7. I am more interested to know how and if the Hungarian Pharmacy Education prepares the pharmacists to get involved in medication review, pharmaceutical care planning processes and to identify potential DDIs rather than reading about the definitions of interventions, etc. I dont find Table 4 , for instance, of a perticular benefit to the readers.

a. In our view, Table 4 is needed to clarify interventions.

b. Line 81: „In Hungary, pharmacists receive a degree after five years at university. In the first two years, basic science knowledge (mathematics, chemistry, biology, botany, etc.) is taught to students, while in the second and third years, they study basic medical knowledge (cell biology, biochemistry, physiology, etc.). This knowledge is the basis of pharmaceutical subjects in the 4th and 5th years, including the study of pharmaceutical care as a separate subject for one semester, during which they get acquainted with the most important therapeutic situations in community pharmacies (antibiotic use, bandages, asthma care, diabetes care, etc.), mainly with theoretical education.”

c. Line 406: “Based on the results of the presented study, pharmacy students need to be purposefully prepared to solve DRPs already at the undergraduate level. In the current Hungarian pharmacy practice, the lack of problem-based educational methodology training may cause that pharmacists consult only about the DPRs to the patients and not to use interventions modifying drug therapy in their own competencies. As a first step, the authors of the study introduced the used project methodology into the curriculum reform of Semmelweis University (e.g. classification and recognition of DRPs according to detected situations) but national improvements are needed in the future.”

8. How did you ensure that the patients who were on 5 or medications actually took all of them? How concordance was evaluated, if it was at all?

a. We did not use specific tools to evaluate concordance. The existence of non-adherence as the underlying cause of DRP3 was determined by pharmacists in their judgment based on experience gained during consultations.

9. Lines 127 & 128 : What do you mean by "pharmacists recorded the active substances involved..."?

a. Line 163: „During the medication review, in addition to the fact of interaction risks, pharmacists wrote down the active substances involved in the particular interactions and recorded the pharmacist intervention(s).”

10. Table 2: I suppose you mean "collaboration" rather than "cooperation"?

a. Table 2 has been corrected.

11. Table 3: Did you double check the grouping of the interactions from a clinical risk perspective with other resources such as Stockly's Drug Interactions?

a. We only used UpToDate Lexiomp® to group the interactions from a clinical risk perspective.

12. Table 5: Can you double check the accuracy of the information re "The average number of products +- S.D?

a. The accuracy of calculations has been checked.

13. Does product mean medicine?

a. The use of the terms was confusing, so we improved the manuscript. In this version, we use the term „medicine” instead of „product”.

Results:

14. What are the definitions of "Quantitative Safety problem" and " Non-quantitative safety problem? Are these standard terminologies? if yes, please define them, first.

a. Quantitative safety problems and Non-quantitative safety problems are terminologies used by the authors of the classification used. A medicine presents a safety problem when it causes or worsens a health problem. A DRP is to be considered quantitative when it depends on the magnitude of an effect, while it is considered as non-quantitative when it is drug-related so it does not depend on the magnitude of an effect.

b. Line 152: „*Non-quantitative DRP: the DRP is drug-related, it does not depend on the magnitude of an effect. **Quantitative DRP: The DRP depends on the magnitude of an effect.”

15. Product, Medicine, drugs, active ingredients, active substances are the terms that need to be clearly defined and consistantly used throughout the manuscript. In its current form, the terms are confusing.

a. The use of the terms was confusing, so we improved the manuscript. In this version, we use the term „medicine „or „drug” instead of „product”. The term „active substance” refers to a chemical substance that has a therapeutic effect in the particular medicine.

16. Overall, you have focused on "What" you found! For an international readers, the "Hows" of addressing "What" was found is more interesting/educational. In the same light, you have only written 4 sentences on the analysis of pharmacist intervetions to slove interaction risks?!

a. Line 191: „During the project, pharmacists used a table summarizing the full medication of the patient, which could indicate the current problem (eg interaction and its severity, and related warning), in addition, pharmacists provided patients with written leaflets summarizing the general rules of medication and placed a counseling poster in pharmacies.”

b. Line 318: “Based on pharmacist reports GPs were notified mainly by telephone or in-person, pharmacists did not prefer written contact.”

c. Line 321: “In the education of the patients, the information materials developed for the project were used by the pharmacists.”

17. I wishfully think that you could also run some simple regression analysis to investigate someassociations between patients characteristics and the types/numbers/severity of DDI or DRPs. In its current form, all of the statistical analyses have been purely descriptive.

a. Methods, Results, and Discussion section have been completed with statistical analysis requested.

b. Line 182: „The frequency of interaction risks considered to be clinically relevant was compared by two demographic aspects (gender, age). We examined the incidence of Grade C, D, or X interaction risks separately in men and women, and in patients under or more than 65 years. In addition, the combined incidence of Grade C, D, and X interaction risks was also examined for both aspects (gender, age).”

c. Line 199: “Based on the descriptive statistics, in the case of Grade C, D, or X interaction risks (separately, and together), the relationship between "one or more interactions" versus "no interactions" by gender and age was examined by the chi-square test. The significance level was 5%.”

d. Line 304: „In comparative studies by gender, the incidence of Grade C, D, or X interaction risks was examined separately and then aggregated (C + D + X). No significant difference was found between men and women in either case (p>0.05). Examining the age groups, we found that there was a more frequent Grade C interaction risk in the age group 65 years or older, with a significant difference (p=0.05), while no significant difference was found for Grade D or X interaction risks. Looking at the combined incidence of Grade C, D, and X interaction risks, it can be assumed in professional practice that the older (≥ 65 years) age group is more likely to have clinically relevant interaction risks than the younger age group (p=0.076, close to the significance limit).”

e. Line 348: “Based on the results of the comparative statistical analysis, it can be assumed that the older patients (≥65 years) are more likely to have clinically relevant interaction risks than the younger age group (<65 years). This difference is mainly due to the higher incidence of Grade C interaction risks requiring therapy monitoring, maybe because the number of medications taken is higher in older patients [1], so it has been shown that patients over 65 years require more special attention during the medication review at a community pharmacy. Monitoring of therapy can be solved within the limits of pharmacist competence, and only in the case of manifested problems it is necessary to consult the GP. This confirmed the need for medication review by pharmacists, as the accumulation of Grade C interaction risks in the elderly can be attributed to the current lack of this service and/or the current lack of a uniform procedure for effective pharmacist-GP collaboration. “

Discussion:

18. Line 272: 531 interaction risks per patient? I take this figure with a pinch of salt! This is too high to me and if it's true it needs to be discussed in details.

a. The sentence is incorrect, 0.7 interaction risks per patient were found in the study.

b. The discussion section was completely rewritten.

19. Line 280: It feels that you are pointing finger at the lack of clear communications between patients and physicians, only! I don't think this would be of any help to anyone. Pharmacists in the communities are meant to fill in that gap. So, if the gap is not filled, then the community pharmacists do need to be included in the discussion.

a. The discussion section was completely rewritten.

20. Line 287: Instead of 1/3 you could write it in full i.e., one third...

a. The discussion section was completely rewritten.

b. Line 317: “Pharmacists notified the GP about the problem in 28.4% of cases, compared with nearly two-thirds (63.1%) solved the problem without the GP’s involvement.”

c. Line 364: „In this research, in one-third of cases, OTCs or other products (e.g. dietary supplements) caused interaction risk.”

21. Line 298: To reiterate my ealier point on the need to discuss the current pharmacy education and practices in Hungary in much more details.

a. Line 81: „In Hungary, pharmacists receive a degree after five years at university. In the first two years, basic science knowledge (mathematics, chemistry, biology, botany, etc.) is taught to students, while in the second and third years, they study basic medical knowledge (cell biology, biochemistry, physiology, etc.). This knowledge is the basis of pharmaceutical subjects in the 4th and 5th years, including the study of pharmaceutical care as a separate subject for one semester, during which they get acquainted with the most important therapeutic situations in community pharmacies (antibiotic use, bandages, asthma care, diabetes care, etc.), mainly with theoretical education. 

The health care institutions that cover Hungary most evenly are the community pharmacies operating as part of the primary care. The majority of patients visit pharmacies for two reasons: 1) to get a drug prescribed by a general practitioner or a specialist; 2) to seek advice on relieving their mild symptoms. During a consultation, pharmacists dispense the prescribed drug or recommend an over-the-counter (OTC) medication for the patient's symptoms.”

22. Line 308: Table 6 in thediscussion? What does it add to the discussion that is relevant to the research findings?

a. Table 6 has been moved to the “Results” section.

b. Line 292: „The Grade D or X interaction pairs of these three agents are shown in Table 6.

23. To appreciate Figure 7 better, one would need to know about the current practices and how Figure 7 is better if at all.

a. The discussion section was completely rewritten.

b. Line 415: “In many cases, the rights of the two disciplines (GPs and pharmacists) are not clearly defined in Hungary in solving the interaction risks, so they need to be clearly defined and widely introduced into the pharmacist practice, e.g. what interaction risks should be indicated to the GP in each case (Fig 7).”

Yours sincerely,

András Szilvay

Corresponding author

---

## [Decision Letter · Decision Letter 1]

10 Jun 2021

Analysis of interaction risks of patients with polypharmacy and the pharmacist interventions performed to solve them – a multicenter descriptive study according to medication reviews in Hungarian community pharmacies

PONE-D-21-08091R1

Dear Dr. Szilvay,

We’re pleased to inform you that your manuscript has been judged scientifically suitable for publication and will be formally accepted for publication once it meets all outstanding technical requirements.

Kind regards,

Vijayaprakash Suppiah, PhD

Academic Editor

PLOS ONE

Reviewers' comments:

Reviewer's Responses to Questions

**Comments to the Author**

1. If the authors have adequately addressed your comments raised in a previous round of review and you feel that this manuscript is now acceptable for publication, you may indicate that here to bypass the “Comments to the Author” section, enter your conflict of interest statement in the “Confidential to Editor” section, and submit your "Accept" recommendation.

Reviewer #4: All comments have been addressed

Reviewer #5: All comments have been addressed

2. Is the manuscript technically sound, and do the data support the conclusions?

Reviewer #4: Yes

Reviewer #5: Partly

3. Has the statistical analysis been performed appropriately and rigorously? 

Reviewer #4: Yes

Reviewer #5: Yes

4. Have the authors made all data underlying the findings in their manuscript fully available?

Reviewer #4: Yes

Reviewer #5: Yes

5. Is the manuscript presented in an intelligible fashion and written in standard English?

Reviewer #4: Yes

Reviewer #5: Yes

6. Review Comments to the Author

Reviewer #4: all comments were justified efficiently

Great effort and great job.

Thanks for all authors for the great effort and i think this paper will add value to the pharmacy practice

Reviewer #5: (No Response)

7. PLOS authors have the option to publish the peer review history of their article (what does this mean?). If published, this will include your full peer review and any attached files.

Reviewer #4: **Yes: **Abdullah Salah Alanazi

Reviewer #5: **Yes: **Keivan Ahmadi

---

## [Editor Report · Acceptance letter]

14 Jun 2021

PONE-D-21-08091R1 

Analysis of interaction risks of patients with polypharmacy and the pharmacist interventions performed to solve them – a multicenter descriptive study according to medication reviews in Hungarian community pharmacies. 

Dear Dr. Szilvay:

I'm pleased to inform you that your manuscript has been deemed suitable for publication in PLOS ONE. Congratulations! Your manuscript is now with our production department. 

Kind regards, 

on behalf of

Dr. Vijayaprakash Suppiah 

Academic Editor

PLOS ONE